# Use of Mepolizumab in Children and Adolescents with Asthma in the USA

**Jaclyn Davis** [1]**, Pamela M. McMahon** [2]**, Emily Welch** [2]**, Cheryl McMahill-Walraven** [3]**, Aziza Jamal-Allial** [4]**, Tancy Zhang** [2]**, Christine Draper** [2]**, Anne Marie Kline** [3]**, Leslie Koerner** [4]**, Jeffrey S. Brown** [2]**, Melissa Van Dyke** [5] **and Ann Chen Wu** [1,2,]*

[1] Pulmonary Division, Boston Children's Hospital, Boston, MA 02115, USA;
  jaclyn.davis@childrens.harvard.edu
[2] Department of Population Medicine, Harvard Medical School and Harvard Pilgrim Health Care Institute,
  Boston, MA 02115, USA; pamela_mcmahon@harvardpilgrim.org (P.M.M.);
  emily_welch@harvardpilgrim.org (E.W.); tczhang@bu.edu (T.Z.); christine_draper@harvardpilgrim.org (C.D.);
  jeff_brown@harvardpilgrim.org (J.S.B.)
[3] Healthagen, Part of the CVS Health Family of Companies, Blue Bell, PA 19422, USA;
  cheryl.walraven@cvshealth.com (C.M.-W.); anne.kline@healthagen.com (A.M.K.)
[4] HealthCore, Inc., Wilmington, DE 19801, USA; ajamal-allial@healthcore.com (A.J.-A.);
  lesliekoerner@yahoo.com (L.K.)
[5] Respiratory Epidemiology Therapy Area, GlaxoSmithKline, Collegeville, PA 19426, USA;
  melissa.k.van-dyke@gsk.com
* Correspondence: ann.wu@childrens.harvard.edu; Tel.: +1-617-867-4823

**Abstract:** Introduction: Pediatric asthma affects 5.5 million US children and is the leading cause of pediatric chronic illness globally. Those with severe asthma have significantly higher healthcare costs compared to those with non-severe disease. Biologics are the newest class of anti-asthma therapy approved for use in patients with severe asthma > 6 years with the eosinophilic phenotype. Objective: The goals of this study were to (1) describe the baseline characteristics of new US pediatric mepolizumab users between 2015 and end dates that varied by data partner (6/30/19–5/31/21), (2) describe asthma medication use in the 12 months preceding and following mepolizumab initiation in this group and (3) assess adherence and persistence to mepolizumab in the 12 months following initiation. Methods: Through an observational cohort study using insurance claim databases, we studied children with a diagnosis of asthma in the preceding 12 months who started mepolizumab and had 12 months of follow-up data. Results: Baseline characteristics of the 72 children who initiated mepolizumab showed variable comorbidities, the most common being allergic rhinitis (88%) and recurrent respiratory infections (71%), as well as varied medication dispensings and patterns of healthcare utilization prior to initiating mepolizumab. Half met the criteria for severe asthma per the GINA guidelines. Comparing weighted averages of treatments dispensed in the 12 months prior to versus following mepolizumab initiation, we observed no significant change in asthma treatments dispensed. Conclusion: This study demonstrates that pediatric patients prescribed mepolizumab have variable previous treatment history and severity of disease, and we found no evidence that mepolizumab alters other asthma medications dispensed in the first 12 months following initiation.

**Keywords:** asthma; pediatric; mepolizumab; biologics

## 1. Introduction

Asthma is the most common chronic condition globally among children, [1] with an estimated 5.5 million cases in the United States affecting 7.5% of all US children [2]. Severe asthma, defined as asthma symptoms that require maximal therapy to remain controlled or that are uncontrolled despite adherence with maximal therapy according to the Global Initiative for Asthma (GINA), occurs in approximately 5–10% of the total asthma population [3]. There is a significant burden associated with childhood asthma, [2] and

severe asthma in particular requires healthcare costs twice as high as those needed to treat mild or moderate asthma [4].

Individuals with severe asthma are a heterogenous group, with variable clinical phenotypes caused by diverse pathophysiologic processes [5]. Classically, severe asthma is considered an eosinophilic-driven process; however, recent efforts have advanced our understanding of the underlying mechanisms to include a less common and poorly characterized neutrophilic phenotype [5]. As the predominant phenotype in children is allergic, there is opportunity for more targeted asthma treatment with biologic therapies in these patients. We do not currently have clear criteria to identify for which patients biologic treatments will be most effective.

Mepolizumab is an interleukin-5 (IL-5) antagonist monoclonal antibody approved by the US FDA in 2015 for add-on maintenance treatment of severe asthma with an eosinophilic phenotype, with current approval for those aged ≥6 years. Its clinical efficacy was evaluated in multiple randomized, double-blind clinical trials that included children >12 years, demonstrating a reduction in the number of exacerbations [6,7] and an improvement in patient-reported quality of life [8]. A more recent pharmacokinetic study in patients age 6–11 years demonstrated safety and tolerance in this population [9]. Studies of mepolizumab's use in real-life populations remain limited. The goal of the present work is to describe the prescribing patterns of mepolizumab among pediatric patients in the US, as well as to characterize medication adherence and persistence in the first 12 months following mepolizumab initiation.

## 2. Methods

### 2.1. Study Design

We conducted a multi-site observational cohort study utilizing administrative claims data, employing methods previously described by our group in a complementary study of mepolizumab use in the adult asthma population [10]. This work leveraged the FDA Sentinel System common data model and distributed analytic tools to facilitate the distributed analyses across health plan partners, which included Aetna, a CVS Health Company, HealthCore, Inc. (Anthem, Inc. data, Wilmington, DE, USA), Harvard Pilgrim Health Care Institute, and the IBM MarketScan® Commercial Claims and Medicare Research [11]. The Harvard Pilgrim Health Care Institutional Review Board determined that this project does not meet the definition of human subject research under the purview of the IRB according to federal regulations.

### 2.2. Study Population

In the present study, we studied children and adolescents with asthma who were <18 years old and initiated mepolizumab after 4 November 2015 and had continuous medical and drug coverage in both the 365 days prior to and following mepolizumab initiation, allowing for a 45-day enrollment gap; our data had end dates of the post-initiation period that varied by data partner (30 June 2019 to 31 May 2021). The 4 data partners had a combined population size (all ages) of more than 114 million individuals. Children and adolescents (<18 years of age) were included if they had a diagnosis of asthma by the International Classification of Diseases (ICD), Ninth Revision, Clinical Modification (ICD-9-CM 493.xx), or Tenth Revision (ICD-10 J45.xx), in the 365 days prior to mepolizumab initiation and had continuous medical and drug coverage in the 365 days prior to and following mepolizumab initiation, allowing for a 45-day enrollment gap. Patients were excluded if they had a diagnosis of cystic fibrosis during any point in enrollment history. This cohort was then followed from the day of mepolizumab initiation until health plan disenrollment, study end date, end of data availability, or death.

### 2.3. Outcome Measures

We examined baseline demographic and clinical characteristics of this population, including age, sex, and comorbid diagnoses. Per GINA guidelines, children and adolescents were classified with severe asthma if in the 90 days before mepolizumab initiation they received:

either medium or high dose ICS with LABA separately or as a combination product; high-dose ICS with LTRA; tiotropium; ≥28 days of systemic steroids; omalizumab; benralizumab; or dupilumab. Healthcare utilization for severe asthma exacerbation was studied by measuring asthma-related hospitalizations, ED visits or ambulatory visits, or need for oral corticosteroid medication for 3–27 days two weeks before or after an outpatient asthma visit.

We examined weighted averages of medication category use in the 12-month period preceding and following initiation of mepolizumab. Medication categories included: short (3–27 days) and long (≥28 days) course OCS; low/medium-dose ICS; high-dose ICS; short-acting beta-agonist (SABA); LAMA; LTRA; ICS (any dose)/LABA; ICS (low/medium dose)/LABA; ICS (high dose)/LABA; ICS (low/medium dose)/LTRA; ICS (high dose)/LTRA; omalizumab; reslizumab; benralizumab; and dupilumab. We used unpaired t-tests to evaluate differences in medication use pre- and post- mepolizumab initiation.

We measured adherence to mepolizumab with percent days covered (PDC) by dividing the days supplied by 366 days, using an index date based on the first dispensing of mepolizumab. For the PDC calculation to be valid, we required 365 days of continuous enrollment (allowing a 45-day enrollment gap) and ended patients' follow-up on day 366. We assessed early-stage persistence with refill data, defined as a second fill within 29–57 days (4–8 weeks) and a third fill within 29–169 days (4–24 weeks) of the first fill.

## 3. Results

We identified 72 children and adolescents with mepolizumab initiation between 4 November 2015 and 31 May 2021 who met enrollment criteria. Mean age was 14.8 years, and 46% were female. Race was unknown. Descriptive data including baseline and clinical characteristics are provided in Table 1. In the 12 months before mepolizumab initiation, 49% had received inhaled corticosteroids (ICS), 78% ICS/long acting beta-agonist (LABA), 69% leukotriene antagonists (LTRA), 22% long-acting muscarinic antagonist (LAMA), 19% omalizumab, 1% dupilumab, and 0% reslizumab and benralizumab. In the previous 12 months, 88% had a comorbid diagnosis of allergic rhinitis, 71% had respiratory infections, 49% had sinusitis, and 24% had atopic dermatitis.

We examined asthma-related medication dispensings in the 12 months prior to and following initiation of mepolizumab. Prior to starting mepolizumab, 50% had at least one dispensing of a medication indicating severe asthma, as defined according to GINA guidelines (either medium or high dose ICS with LABA separately or as a combination product, ≥28 days of oral corticosteroids (OCS), high-dose ICS with LTRA, omalizumab, dupilumab, or tiotropium). Additionally, 85% of those studied utilized short courses of OCS during this time frame. Weighted averages of medication category use in the 12-month period preceding and following initiation of mepolizumab are summarized in Figure 1. We did not find evidence of significant difference at the 5% significance level in medication use pre- and post-mepolizumab initiation.

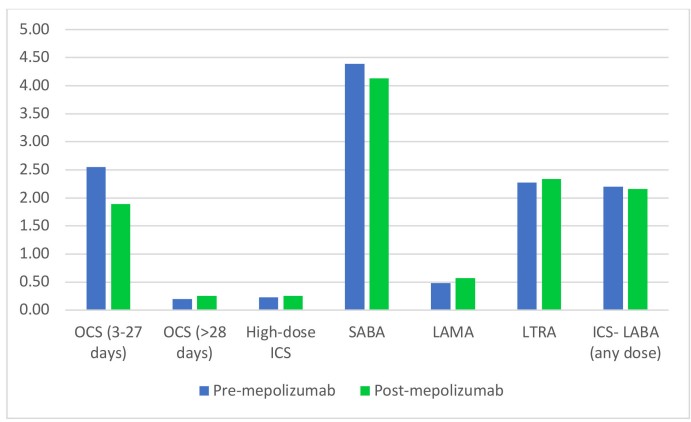

**Figure 1.** Mean weighted average medication dispensing before and after mepolizumab initiation. Significant differences in dispensation patterns were not detected pre- and post- intervention.

**Table 1.** Baseline demographic and clinical characteristics of new mepolizumab users <18 y.

| Characteristic | N/Mean | %/SD [1] |
|---|---|---|
| Number of unique patients | 72 | |
| **Demographics** | | |
| Mean Age | 14.8 | 2.0 |
| Gender (Female) | 33 | 45.8% |
| **Recorded history of:** | | |
| Allergic rhinitis | 63 | 87.5% |
| Respiratory infections | 51 | 70.8% |
| Sinusitis (acute/chronic) | 35 | 48.6% |
| Atopic dermatitis | 19 | 23.6% |
| Eosinophilia | 21 | 26.4% |
| Eosinophilic esophagitis | 13 | 18.1% |
| Nasal polyps | 6 | 8.3% |
| Chronic obstructive pulmonary disease | 4 | 5.6% |
| Eosinophilic granulomatosis with polyangiitis | 2 | 2.8% |
| **History of use:** | | |
| ICS | 35 | 48.6% |
| ICS and LABA | 56 | 77.8% |
| ICS and LTRA | 20 | 27.8% |
| LAMA | 16 | 22.2% |
| Omalizumab | 14 | 19.4% |
| Dupilumab | 1 | 1.4% |
| SABA | 68 | 94.4% |
| OCS | 61 | 84.7% |

[1] Value represents standard deviation where no % follows the value.

With regards to severe asthma exacerbations in the 12 months prior to initiating mepolizumab, 78% had the combination of at least one outpatient asthma diagnosis and 3–27 days of OCS supplied, 51% had an ambulatory asthma-related event, and 21% experienced a hospital admission for asthma. In the 12 months following mepolizumab initiation, 63% had at least one outpatient asthma diagnosis and 3–27 days of OCS supplied, 50% had an ambulatory asthma-related event, and 13% experienced hospital admission for asthma. These rates of severe asthma exacerbations were not statistically different pre- and post-initiation.

We also analyzed adherence to mepolizumab in this population, including PDC and discontinuation patterns; early-stage persistence was 63.9%.

## 4. Discussion

Our report is the first to describe baseline characteristics of pediatric patients in the US treated with the biologic asthma therapy, mepolizumab. This real-world study demonstrates mepolizumab is used in children and adolescents with asthma who have a range of co-morbid illnesses and asthma medication history. The most common co-morbid diagnoses were allergic rhinitis and recurrent respiratory infections. These co-existing conditions are unsurprising in this population, as allergic rhinitis is associated with worse symptom control in those with asthma [12], and recurrent respiratory infections are commonly associated with asthma exacerbations. In our study, children and adolescents who initiated mepolizumab demonstrated relatively high adherence to mepolizumab in

the first year. We did not detect a difference in severe asthma exacerbations or additional asthma medication dispensings in this small cohort when comparing pre- versus post-initiation time periods.

Though prescribing guidelines for mepolizumab include a diagnosis of severe asthma, only half of those studied met GINA criteria for severe asthma by controller medication requirement. This may suggest that prescribers are prescribing this biologic in those for whom there is room for step-up therapy before reaching for a biologic. There are several possible explanations for this. One is that prescribers may have a poor understanding of the indications for mepolizumab and may be inappropriately prescribing this medication to a segment of the population for whom it is not indicated. Alternatively, prescribers may be intentionally prescribing biologics to this population due to beliefs in improved adherence with intermittent biologics versus daily or twice-daily treatments [13].

Another possibility is that prescribers may be rightfully classifying their patients as having severe asthma by criteria other than those put forth in the GINA guidelines. The National Asthma Education and Prevention Program (NAEPP) [14] classification for severe asthma is based on symptoms and lung function rather than maximal controller treatment. It is plausible that more of our studied children and adolescents may have met severe asthma criteria under this classification. This idea is supported by our study's finding that 84% of our studied population required OCS treatment, indicating uncontrolled asthma, in the year leading up to mepolizumab initiation. This cohort may have also demonstrated high symptom burden, impaired lung function, and other markers of airway inflammation, which were unable to be captured as part of this study design. It is important to understand who comprises the population of pediatric patients receiving biologics for severe asthma as well as their response.

Strengths of this report include its novel real-world description of new pediatric mepolizumab users in the US as well as its ability to accurately measure rates of severe asthma by medication use via claims database analyses. There are also limitations to acknowledge. This study utilized claims data from private insurers, and this may limit generalizability. Our study period included the COVID-19 pandemic, which was associated with decreased rates of asthma exacerbations; [15] this may also limit generalizability. We recognize claims data define medications dispensed and may not capture actual medication usage for medications other than the biologics, which were captured accurately as part of a required medical encounter. Nevertheless, medications dispensed is a more accurate representation of actual medication usage than medications prescribed. While this cohort includes only patients who have an asthma diagnosis, we do not know the exact prescribing indication for mepolizumab. Statistical testing was limited by the use of aggregated data and inability to perform paired analysis. Importantly, our data did not include clinical measures of asthma impairment (e.g., symptoms, lung function testing, fractional excretion of nitric oxide (FeNO)) or laboratory values such as IgE or eosinophil counts, which may contribute to the appropriate prescribing of biologics for patients with asthma and could in part explain the low rates of severe asthma identified in this patient population. Additionally, because data were derived from insurance claims, certain diagnoses or conditions associated with this population, such as COPD, may not be accurate and are difficult to verify.

In summary, mepolizumab is being used in children with asthma with and without severe asthma, as classified by well-accepted international criteria. Further data are needed to understand mepolizumab's impact on clinical outcomes, including asthma symptom scores, hospitalization rates, missed school days, and quality of life, as well as to develop ways to identify for which patients this therapy is most effective. This knowledge will facilitate a better understanding of the therapy's cost effectiveness and may allow for more precise use of this costly asthma treatment.

**Author Contributions:** Conceptualization, J.S.B., M.V.D. and A.C.W.; data curation, P.M.M., E.W., C.M.-W., A.J.-A., T.Z., A.M.K., L.K. and A.C.W.; formal analysis, P.M.M., E.W., C.M.-W., A.J.-A., T.Z., A.M.K., L.K. and A.C.W.; funding acquisition, J.S.B., M.V.D. and A.C.W.; investigation, A.C.W.; methodology, P.M.M. and A.C.W.; project administration, P.M.M., C.D. and A.C.W.; supervision, J.S.B., M.V.D. and A.C.W.; validation, A.C.W.; visualization, P.M.M.; writing—original draft, J.D., P.M.M. and A.C.W.; writing—review and editing, P.M.M., E.W., C.M.-W., A.J.-A., T.Z., C.D., A.M.K., L.K., J.S.B., M.V.D. and A.C.W. All authors have read and agreed to the published version of the manuscript.

**Funding:** This research was funded by GSK (PRJ2757).

**Institutional Review Board Statement:** The Harvard Pilgrim Health Care Institutional Review Board determined that this project does not meet the definition of human subject research under the purview of the IRB according to federal regulations.

**Informed Consent Statement:** Patient consent was waived given that data were aggregated into a large de-identified dataset, with no way to identify who the patients were.

**Data Availability Statement:** The data presented in this study are available on request from the corresponding author.

**Conflicts of Interest:** This study was funded by GSK. Melissa Van Dyke is an employee and shareholder of GSK. The authors have no additional conflict of interest.

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
