# Peer review of "Use of Mepolizumab in Children and Adolescents with Asthma in the USA"

_2673-527X, doi:10.3390/jor2030010_

Round 1
Reviewer 1 Report
The article is clear and well written. It provides an interesting insight of the usage of Mepolizumab on pediatric patients with severe asthma in the US. Maybe it would have been interesting to put up a comparison between European paediatric patients, so as to investigate if the findings are shared with other countries.
Author Response
Thank you for this comment. Unfortunately we are limited to US claims data and are not positioned to make a comparison to European patients.
Reviewer 2 Report
The paper „The use of mepolizumab in children and adolescents with asthma in the US” by Davis et al. shows how this treatment option is used in a group of patients in the US. As said in the abstract, patients receiving mepolizumab did not have a similar history of previous treatment and that there is no evidence for a profit in the use of this medication.
In the introduction, the authors state that it is not known in which patients biologics might be most effective. In fact, please have a look on https://www.ncbi.nlm.nih.gov/pmc/articles/PMC7606967/ and consider to discuss this issue in the discussion.
In the results the authors show that there were no significant changes in the use of other medications and number of asthma exacerbations before and after treatment even if the main goal of the use of biologics is alleviation of asthma symptoms, decrease of use of medications and improved quality of life. What is the reason for such an observation? Where the real reasons or results missing because of the character of the data (from insurance sources)? The authors mention that the main reason for unsuccessful use of the medication are prescriptions in non-severe asthma. Why could be the reason for that? The belief that “new” medications are “better” than the “old” ones or that “biologic” means “more natural” or “safer”?
Why did the authors include into the study subjects with asthma/COPD overlap syndrome and eosinophilic granulomatosis with polyangiitis as these diseases are rather rare in children? Could the authors comment on this?
Please check formatting in the entire document
